# Advances in the Measurement of Polymeric Colorimetric Sensors Using Portable Instrumentation: Testing the Light Influence

**DOI:** 10.3390/polym14204285

**Published:** 2022-10-12

**Authors:** Adria Martínez-Aviño, Maria de Diego-Llorente-Luque, Carmen Molins-Legua, Pilar Campíns-Falcó

**Affiliations:** MINTOTA Research Group, Departament de Química Analítica, Facultat de Química, Universitat de València, Dr. Moliner 50, 46100 Burjassot, Valencia, Spain

**Keywords:** polymeric membranes, PDMS, cellulose, color measurement, RGB, smartphone, light

## Abstract

Sustainable and green sensors based on polydimethyl siloxane (PDMS) or cellulose polymers, as a case of study of the use of portable instrumentation joined to a smartphone, have been tested. A smartphone camera was used to obtain images and was also coupled to a minispectrometer, without and with an optical fiber probe to register spectra. To study light influence on the analytical signal, light-emitting diode (LED), halogen light and daylight have been assayed. A corrective palette of 24 colors and a set with 45 colors from different color ranges were used as the validation set. The results indicated that halogen light was the best option to obtain the spectra. However, for digital image analysis, it was the LED light that gave a greater approximation of the RGB values of the real colors. Based on these results, the spectra and the RGB components of PDMS solid sensors doped with 1,2-naphtoquinone-4-sulfonate (NQS) for the determination of ammonium in water or urea in urine, PDMS doped with Griess reagent for developing the assay of nitrite in waters and cellulose sensors for the determination of hydrogen sulfide in the atmospheres have been obtained. The results achieved were good in terms of sensitivity and linearity and were comparable to those obtained using a laboratory benchtop instrument. Several rules for selecting the most suitable light source to obtain the spectra and/or images have been established and an image correction method has been introduced.

## 1. Introduction

In situ analysis is a trending topic in the field of analytical chemistry. One option proposed to address this challenge is to use sensors combined with portable instrumentation [1]. In this context, Campíns-Falcó et al. proposed the use of polymers as supporting materials to embed chromogenic [1] or fluorescent [2] reagents. Between these materials, polydimethyl siloxane (PDMS) stands out. PDMS is highlighted for its properties, such as good thermal stability, biocompatibility, flexibility, low cost, easy to use, chemical inertness, transparency and gas permeability. While it has a low modulus of elasticity and strength, these can be improved by mixing PDMS with other polymers or by adding particles, such as SiO_2_ nanoparticles (NPs). To reduce the hydrophobicity of the material, tetraethyl orthosilicate (TEOS) or liquid ionic have been employed [3,4]. The reagents entrapped in the PDMS become very stable over time and they can be kept in the solid sensor or delivered to the solution where the assay will be carried out. Other polymeric solid sensors based on cellulose have been proposed [5]. This material presents high porosity, flexibility, versatility, biocompatibility, and non-toxicity. These sensors have been manufactured based on the principles of sustainability and greenness, with a low consumption of energy and reagents, low toxicity, and low generation of waste, among other things. The permeability of these materials allows changes in color to be measured by reflectance, by the transmittance mode or even from images. Additionally, these analytical signals can be readout using portable instrumentation.

Due to the interest in in situ analysis, portable instrumentation has become a subject of growing interest in the last years in environmental [6], health [7] and food monitoring [3], among other areas. Portable devices allow for the performing of in situ analysis, saving a lot of time in sample collection and facilitating the decision-making. In addition, portable instrumentation is versatile, cost-effective and easy to manage.

Traditionally, colorimetric analysis is a common technique for the evaluation of organic and inorganic compounds. This can be conducted by the naked eye for qualitative or semiquantitative analysis [3,8], or by a noninvasive techniques, such as lab-visible reflectance spectroscopy for quantitative analysis [3]. However, new portable instruments with different configurations and possibilities, such as optical fiber probes and miniaturized spectrometers, have recently been developed [9,10]. At present, smartphones are extensively used worldwide. Therefore, they have become a profitable tool under the umbrella of in situ analysis. Accordingly, a variety of studies have been performed using smartphones as analytical devices [11,12,13,14,15,16,17]. A smartphone can be used as a camera in order to take pictures for color analysis and to obtain color parameters, such as RGB (red/green/blue) coordinates [5] or CIElab parameters [11]. Digitalized images can also be processed by external programs, such as GIMP, imageJ or MATLAB. As can be seen in previous studies, the processing of images can improve the results obtained, achieving higher linearity and sensitivity [2]. In the last decade, digital cameras and smartphones facilitated the emergence and development of new devices for color analysis at more affordable prices [11,18]. In this sense, smartphones with more sophisticated cameras have been launched. New camera settings provide the opportunity to customers to adjust parameters as color temperature, sensitivity to light (ISO), exposure time and contrast ratio, reducing the number of systematic errors in the analysis of color parameters of the image [19]. A color analysis shows a strong dependence on light conditions. Thus, monitoring the incident of light in each measurement is of vital importance. The light intensity entering the camera detector can directly determine the RGB values of each pixel of an image. Image quality can suffer from non-uniformity and non-reproducibility, which negatively affects the accuracy of the measurements. To obtain good measurements, some strategies, such as the use of boxes with light [20] or using the light of the phone, have been proposed to eliminate interference from ambient light [21]. Depending on the type of light, alternative color temperature can be observed, and additionally, different information in certain wavelength ranges can be provided by each type of light. All these features must be taken in consideration prior to adjusting the camera settings to obtain the most representative images. Additionally, some algorithms have been used to obtain more precise images [22,23,24].

In this work, measurements were carried out using a smartphone, combined with a mini-spectrometer that uses its camera and coupled or not to an optical fiber probe for spectra analysis, and a smartphone with a digital camera in order to obtain RGB parameters. Digitalized images were processed using external programs with the aim of improving the results obtained. The influence of different types of light in this portable instrumentation have been evaluated. Three different ambient lights, namely LED, halogen and daylight, were first characterized. These lights have been applied as incident light and have been compared to the results obtained by a laboratory benchtop spectrophotometer.

A protocol guide to perform suitable measurements has been established. It employed a color correction palette of 24 colors and a testing set of 45 colors. Two polymeric colorimetric solid sensors were examined as a case study. The first was used to determine ammonium in water and urea in urine and the second for hydrogen sulfide in the atmosphere. However, the presence of nitrite ions in water was also tested as an example of delivering from polymeric sensor. A comparative study for all types of light for each application has been performed. To conclude, some rules for selecting the most suitable light source for obtaining the needed information have been set up. While many articles have been published on this topic, as far as we know, it is the first time that incident light has been characterized with a portable spectrometer to fix the measurement conditions of the smartphone coupled or not to the mini-spectrometer with or without an optical fiber probe, in order to enhance the analytical results obtained from the image analysis or the spectra, respectively. Additionally, a color correction method has been performed by adjusting the contrast and brightness parameters to achieve more realistic colors in the case of image analysis.

## 2. Materials and Methods

### 2.1. Reagents and Solutions

Ultrapure water obtained using the Nanopure II system (Barnstead, NH, USA) was used for the preparation and dilution of all the solutions. N,N-Dimethyl-p-phenylendiamine dihydrochloride was purchased from Sigma-Aldrich (St. Louis, MO, USA). Hydrochloric acid (37%) and sodium sulfide hydrate were acquired from Scharlau (Barcelona, Spain). Iron (III) chloride hexahydrate was provided from Probus (Barcelona, Spain). Glycerol was obtained from Sigma-Aldrich (USA). Grade 41 Whatman filter papers were used as a support. PDMS membranes were synthesized using a Sylgard^®^ 184 Silicone Elastomer Kit (base and curing agent) obtained from Dow Corning (Midland, MI, USA). Sodium 1,2-naphthoquinone-4-sulfonate (NQS, 99.7%), tetraethyl orthosilicate (TEOS ≥ 99.0%), silicon dioxide nanoparticles (SiO_2_ NPs, 99.5%, 5–15 nm particle size), urease (Canavalia ensiformis–Jack bean 64,347 units/g in 0.31 g), and APTMS (aminopropyltrimethoxysilane) were purchased from Sigma-Aldrich (USA). Sodium carbonate, sodium hydrogen carbonate and ammonium chloride were provided by Probus (Spain). Urea, 2-propanol (≥99.9%) and sodium hydroxide were provided by VWR Chemicals (Radnor, PA, USA). Trichloroacetic acid (≥99.0%) and sodium dihydrogen phosphate monohydrate were obtained from Merck (Darmstadt, Germany). Di-sodium hydrogen phosphate anhydrous was obtained from Panreac (Barcelona, Spain). Sulfanilamide was purchased from Guinama (Valencia, Spain), potassium nitrite was obtained from Merck (Germany) and citric acid monohydrate was obtained from VWR Prolabo (Lavonia, Belgium).

### 2.2. Apparatus and Materials

For the lightning characterization, a handheld tool was used (WaveGo, Wave illumination-Ocean Optics) and controlled by a Smartphone (Xiaomi Redmi Note 8). A white box with available LED light (JZUO, Puluz, Amazon) was used. As incident light, a halogen lamp (20 W) 12V type MR11 was used, and for LED illumination, lighting was switched on from the box light using 2 strips of 35 pieces of white LED light each. For the analytical response measurements, smartphone (Samsung Galaxy A70) and smartphone-mini-spectrometer (GoSpectro, Goyalab), coupled or not to an optical fiber probe, were used. UV–Vis spectra in reflectance mode were also registered with a Cary 60 UV–Vis spectrophotometer (Agilent) equipped with a diffuse reflectance probe from Harric Scientific Products. Two different procedures were used to obtain the RGB components: (i) Non-processed images and (ii) processed images using the GIMP free program.

A corrector Spider Checker Color V2 of 24 colors was used to calibrate the image color and a set of 45 colors, which covered the visible color range, were selected as the validation set (Figure 1a,b).

### 2.3. Polymeric Chemosensors

#### 2.3.1. Paper-Based Sensors for Hydrogen Sulfide

A mixture of 50 μL of a 1:1:0.1 of 0.25 M of FeCl_3_, 0.28 M of N,N-Dimethyl-p-phenylenediamine and glycerol was added to each cellulose paper sensor (1 cm of diameter). After 15 min of vacuum drying, the sensor was exposed to a generated hydrogen sulfide atmosphere. The sensor was left to react for 30 min and then washed with 5 mL of water to remove the excess reagent. Finally, the color response was measured [5] (see Figure 1c).

#### 2.3.2. PDMS-Solid Sensors for Ammonium or Urea

PDMS doped with NQS sensors were synsethized [4]. The reaction was performed by introducing the sensing PDMS membrane into a vial containing 1 mL of Na_2_CO_3_ buffer solution (pH = 11) and 1 mL of ammonium solution. The vial was heated to 100 °C for 10 min. In the presence of ammonia, the PDMS sensors’ color changed. For urea determination, a sample pretreatment was carried out on the urine samples [3]. Urea was measured as ammonia following the procedure described above (see Figure 1c).

#### 2.3.3. PDMS Delivery Sensor for Nitrite

Microplates of 96 wells were used for nitrite determination. A 5 mm of diameter synthetized PDMS doped with Griess reagent was introduced in each well and mixed with 150 μL of 330 mM citric acid and 150 μL of nitrite standard or water sample. The PDMS sensor delivered the reagent to the solution. The microplate was then shacked for t = 10 min at room temperature. Then, measurements were performed [25] (see Figure 1c).

### 2.4. Experimental Design and Data Acquisition

To evaluate the effect of light sources, experiments were performed under controlled illumination conditions. A portable spectrometer (WaveGo) was used to characterize the light. When photos were taken on the smartphone, the camera settings were fixed in auto-focus, ISO at 100 and brightness at 1.3. The color temperature was fixed depending on the light used. The smartphone was situated on top of a box, within 5 cm of the sensor or the image to be photographed (Figure 2a,b). The acquired images from the smartphone were processed using the GIMP program to obtain the RGB values. Images were also obtained inside a folding white box (Appendix A, Figure 3a) at 5 cm. At the same light conditions, the spectra were registered using the smartphone, coupled to a mini-spectrometer (GoSpectro) without (Figure 2a,b) and with fiber optic, as shown in Figure 3b. The fiber optic was situated at 0.5 cm of the color or sensor. The spectra were registered in both the reflection and transmission mode for the NQS transparent sensor (Figure 2b) using the smartphone-GoSpectro. In Figure 1c, images from the three different selected chemosensors, before and after the colorimetric reaction, are shown. Spectra from the benchtop lab instrument were also obtained for the comparative proposal.

## 3. Results and Discussion

### 3.1. Characterizing of Incident Lights

Three lights sources: Halogen lamp, white LEDs and daylight were tested (see Section 2.2). A portable spectrometer (WaveGo) was used to analyze different parameters of incident light, such as light color temperature CCT (K), intensity (LUX), and color rendering index (CRI). CCT(K) is the temperature of an ideal black body radiator that radiates light of a color that is comparable to that of the light source and it is a characteristic of visible light. LUX refers to the strength or amount of light produced by a specific lamp source. This measures the wavelength-weighted power emitted by a light source. Illumination intensity is a physical term that refers to the luminous flux of visible light received per unit area. The unit is Lux or lx, referred to as illuminance. This is used to indicate the intensity of the light and the surface area of the object being illuminated. The CRI value is the measurement of how colors look under a light source when compared to sunlight. This parameter indicates how accurately a color is represented in the measured light versus an ideal light source. Figure 4 shows the spectra and light parameters obtained for the three light sources studied in this work. The lights used had differences in the three parameters mentioned.

### 3.2. Measuring of the Spectra

A first set of experiences consisted of registering the vis spectra of the color correction palette of 24 colors, and a set of 45 colors using the three light sources in order to evaluate the influence of light. We used a lab benchtop instrument for comparative purposes and a smartphone combined with the mini-spectrometer coupled to an optical fiber probe (Figure 3b).

#### 3.2.1. Color Correction Palette

Spectra of reflectance, expressed as absorbance of the different colors (see Figure 2a), were obtained at the three light sources. The parameters of the smartphone camera were set to 1–1.3 brightness contrast, ISO 100, auto-focus mode and color temperature according to the light used (see Figure 4). Focus was mostly kept in the three primary colors, red, green and blue. Similar shapes of the spectra were obtained for all the options, however there were differences in the absorbance intensities, as can be observed in Figure 5. These differences could be related to the light intensity, as the higher the lux value, the higher the absorbance intensity observed and CRI parameter. The pattern of spectra obtained using the mini-spectrometer were comparable to those obtained using the lab benchtop instrument.

#### 3.2.2. Set of 45 Colors

The spectra of the panel of 45 colors were clustered into six groups depending on their color tone (yellow, red, brown, grey, blue and green). Registered spectra were also compared to the results obtained using the lab benchtop spectrophotometer. As an example, Figure 6 shows the obtained spectra of the brown color group.

It was observed that satisfactory spectra were obtained in terms of absorbance when daylight was used as the light source. Registered spectra using a white box with LED illumination showed a suitable absorbance signal and halogen lamp presented spectra with a higher absorbance signal, allowing colors from the same group to be differentiated more efficiently. Good precision was achieved, as can be observed from the results in Table 1, although daylight provided higher percentage rsd values, which can be explained due to the changes that sunlight can suffer throughout intra-day and inter-day analysis. Spectra obtained for red and blue color groups for all light sources are available in Appendix A and the same remarks can be made.

The spectra obtained using a conventional reflectance diffuse instrument was similar to those achieved using the mini-spectrometer (GoSpectro) independently of the light source used. GoSpectro only works in the 420–680 nm range of the wavelengths. Better spectra shapes were obtained using halogen or daylight instead of LED light in reference to those provided by the lab benchtop equipment. Halogen light was selected as the best option to measure the spectra.

### 3.3. Image Analysis and RGB Color Coordinates Registration

A smartphone as analytical instrument for image analysis has many advantages [26], including easy operability, quick readout and good connectivity, among others. However, the smartphone readout is susceptible to the measuring conditions, besides the light conditions by the kind of metal oxide semiconductor (CMOS) of the camera. Many solutions have been published in order to control the light conditions [27]. Nowadays, the camera parameters can be controlled, and fixed conditions can be stablished. To minimize the influence of light and the quality of the camera on the color coordinates, the use of a standard color card for calibration has been studied.

Here, images were taken using the professional mode of the camera of the smartphone and parameters were those indicated in Section 3.2.1 for the coupling smartphone-mini-spectrometer.

Further, the next experiments were focused primarily in the three primary colors, red (14), green (18) and blue (22) of the color correction palette (Figure 1a). A linear correction method was studied to rescale the RGB parameters of the tested colors by adjusting the contrast and brightness values of the image using the GIMP program. These variations were applied to obtain the RGB coordinates of the black and white references as close as possible to pure black (0, 0, 0) and pure white (255, 255, 255), respectively. It was observed that almost pure primary colors were obtained for the processed images, (255, 0, 0) for red, (0, 255, 0) for green and (0, 0, 255) for blue. In any case, independent of the light used, the processing treatment improved the responses of the RGB parameters, and a better approximation to the real RGB components was reached, as can be seen in Figure 7.

When using LED light, image processing removes the G component from red and blue colors and the B component from the red and green colors, thus obtaining more purified primary color, as observed in Figure 8. However, when halogen light is used, image processing removes the R component from blue and to a lesser degree the green color, and intensifies it for red. The same is obtained for daylight.

The range of maximum and minimum values of the RGB parameters, for the original and processed images, was determined for each primary color, using the different light sources. It was observed that the RGB coordinate, which allows for a complete differentiation between colors, is the R parameter as it ranges for the different primary colors were clearly defined. In addition, a better ratio separation for each primary color was observed when the images were processed, within ranges of R_Red_ (224–255), R_Green_ (17–74) and R_Blue_ (0–6). The distribution 3D of the 24 colors of the correction palette according to their RGB parameters using the different light sources by unprocessed and processed images are given in Appendix A.

The same study was applied for the panel of 45 colors. As previously discussed, colors were clustered into the six aforementioned groups depending on their color tone. It was observed that, for the color ranges composed of warm colors, such as yellow, red and brown, halogen and LED light sources provided more information for all the RGB coordinates. Conversely, for the color ranges composed of cold colors like green and blue, only the LED light source achieved good separation in terms of RGB values for colors with similar tonalities. The grey group of colors remained significantly unchanged in regards to light conditions as they are colors in the scale between white and black. Additionally, image processing allowed for a better differentiation between colors in the same group, which meant a higher sensitivity for color analysis. A common trend observed for all color groups was that LED light reported the most useful details for color evaluation. Additionally, this set up caused a lower loss of information when processing images. Depending on the analyzed color, the best results were observed with different RGB coordinates, as its suitability hinges on the color tone. In this sense, the yellow group showed a better differentiation in the B parameter, while red and brown color groups showed that the G component provided more useful information, and for green and blue colors, the R coordinate was the most suitable to perform the color analysis.

### 3.4. Colorimetric Chemosensors as a Case of Study: Scaling the Information

Three different polymeric chemo-sensors for different analytes have been studied using the aforementioned conditions to obtain their analytical responses. The methodology applied previously for the set of 45 colors was also used in these cases.

A PDMS solid sensor doped with NQS for ammonia and urea, PDMS doped with Griess reagent as a delivery sensor for nitrite, and paper-based sensor for hydrogen sulfide were selected as the different cases of study (see Figure 1). Based on the reaction products formed, the wavelength selected for measuring were 490, 670 and 540 nm, respectively. The solid sensors allowed the gas or the liquid to penetrate the material if there was appropriated color intensity and uniformity.

Spectra analysis results proved that, as previously established, a slightly higher sensitivity was observed when using a halogen lamp instead of the LED source. As an example, Figure 9 shows the spectra obtained for the PDMS-NQS sensors using halogen light and transmission (Figure 2b) and reflection modes (Figure 2b and Figure 3b) with the GoSpectro and those obtained with the lab benchtop spectrophotometer. The registers of the GoSpectro, coupled to the fiber optical probe, were more similar to those obtained by the lab instrument, as can be seen in Figure 9.

The figures of merit for the different sensors are shown in Table 2. The limits of detection (LODs) and limits of quantification (LOQs) were calculated as 3s_B_/b_1_ and 10s_B_/b_1_, respectively, where s_B_ and b_1_ are the standard deviations of the blank (n = 3) and the slope of the linear regression. The linear range and sensitivity of the methods were also evaluated. As can be seen in Table 2, no significant differences were obtained between the reflectance or transmittance mode for the NQS-PDMS sensor. The sensitivity obtained using the mini-spectrometer with fiber optic was higher than those achieved by the other measurement approaches. Based on these results, the reflectance mode and the use of fiber optic, coupled to the mini-spectrometer and smartphone, were selected for further experiments. Additionally, suitable figures of merit were obtained for the nitrite assay from delivering the PDMS device and paper-based sensor for hydrogen sulfide, as can be seen in Table 2.

According to the studies carried out above, the RGB coordinates showed that the best results were obtained using the LED light, and good results were observed in terms of sensitivity and linearity in all cases (Table 2). Different RGB parameters were chosen for each sensor, since, as established earlier, depending on the colorimetric reaction product, a better correlation was obtained with a specific coordinate. It was observed that, generally, the best correlation was obtained with the RGB parameter with a closer tone to the complementary color of the analyzed sensor. Thus, the G coordinate was chosen for the ammonium/urea and nitrite sensors, while for the hydrogen sulfide sensor, the best results were achieved with the R parameter. Corrected images, in accordance with the method proposed and described in Section 3.3 for the sensors, were used to establish the figures of merit for the three sensors.

## 4. Conclusions

In this paper, several approaches have been proposed to measure the color provided by the polymeric chemosensors by image analysis and from the spectra. Different light sources, characterized by a portable spectrometer and several measurement methodologies employing a smartphone, and also its coupling with a mini-spectrometer without or with a fiber optical probe, have been evaluated. The aim was to establish some guidelines to select the appropriate set up. LED light, halogen light and daylight sources were tested. A corrector color palette and a set of 45 colors with different color groups were studied in order to establish rules to obtain the optimal responses and to apply them to the selected sensors by using the RGB color coordinates. The colors developed on three polymeric chemo-sensors for different analytes, namely urea/ammonium, hydrogen sulfide and nitrite, have been tested. Figures of merit for several sensors were obtained as linearity, LODs, and LQDs obtained for the spectra analysis were better when they performed the measurements using a halogen lamp, however better results were obtained with the LED light when the RGB coordinate analysis was carried out. Daylight results were suitable in terms of linearity and sensitivity, but not in terms of precision, as its light intensity varies from day to day. In this sense, it has been demonstrated that controlling the measurement conditions is of vital importance to perform suitable analysis using the portable instrumentation tested. The choice of a specific RGB color coordinate was also studied in this work and a correction method for approximating the real color was proposed. It was observed that the best results were obtained when the RGB parameter was closer to the complementary color of the tone of the analyzed sensor. The achieved results indicate that the halogen and LED lights are suitable options to perform in situ analysis, based on the use of a mini-spectrometer coupled to a smartphone or images of the sensors, and their adequacy can be evaluated depending on the color information. Furthermore, image processing has also proven to be an appropriate tool to carry out color analysis.

## Figures and Tables

**Figure 1 polymers-14-04285-f001:**
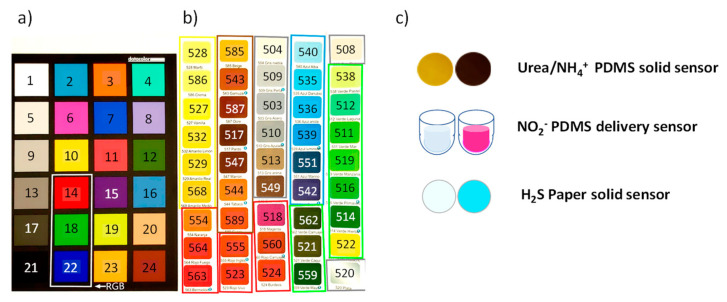
(**a**) Color correction palette of 24 colors; (**b**) set of 45 colors of different spectral regions; (**c**) and blank sensor and sensor after the colorimetric reaction for the different studied supports.

**Figure 2 polymers-14-04285-f002:**
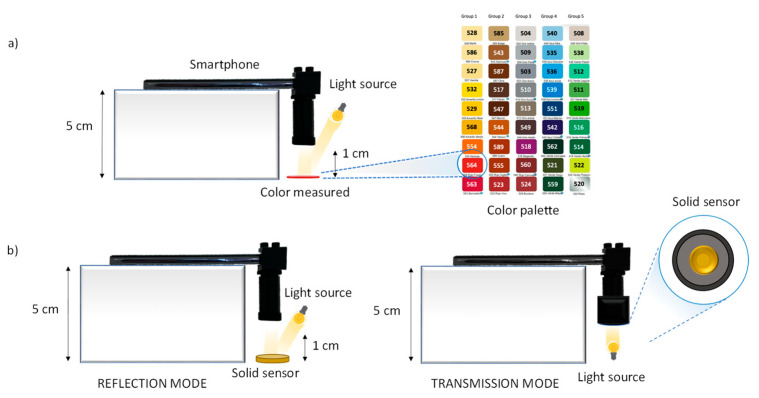
Scheme of the experimental designs; (**a**) smartphone-mini-spectrometer using the transmission mode for the palette of colors; and (**b**) smartphone-mini-spectrometer using reflection, and in particular, the transmission mode for the NQS sensors.

**Figure 3 polymers-14-04285-f003:**
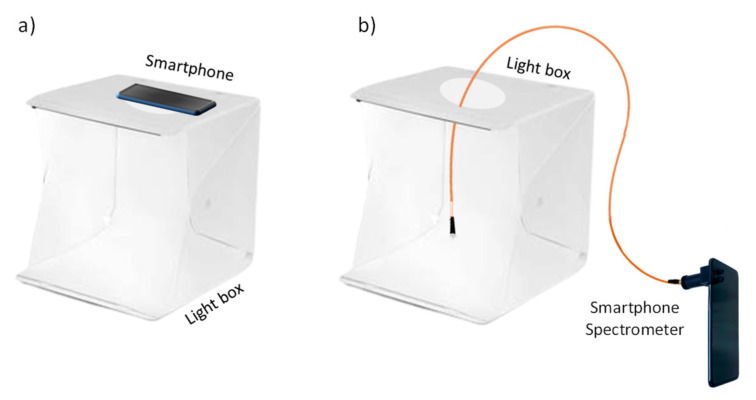
Scheme of the experimental designs for white box measurements; (**a**) RGB smartphone image analysis; and (**b**) smartphone-mini-spectrometer coupled to an optical fiber probe of reflection.

**Figure 4 polymers-14-04285-f004:**
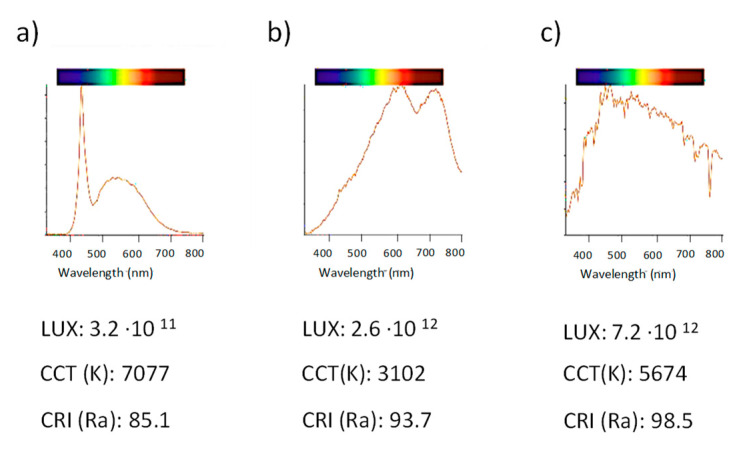
Light spectra and parameters of light for different incident light sources: (**a**) LED light; (**b**) halogen light; (**c**) daylight, using a spectrometer (WaveGo). For further explanations, see the text below.

**Figure 5 polymers-14-04285-f005:**
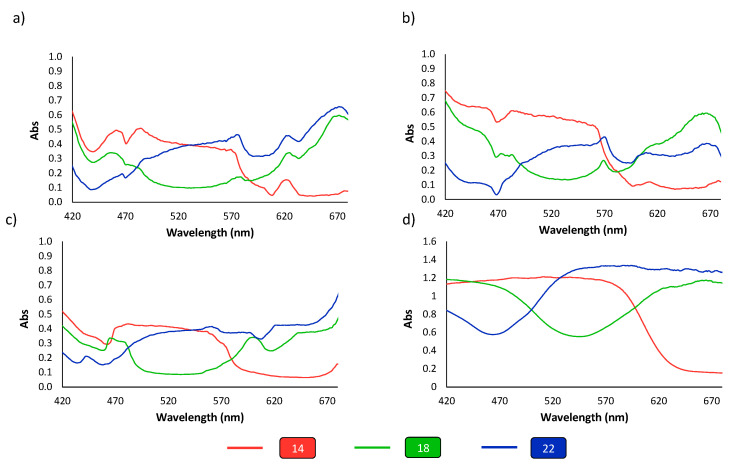
Vis-spectra at different wavelengths of the three primary colors red, green and blue of the color correction palette using the different lights and the mini-spectrometer. (**a**) LED; (**b**) halogen; (**c**) daylight; and by (**d**) a laboratory benchtop spectrometer.

**Figure 6 polymers-14-04285-f006:**
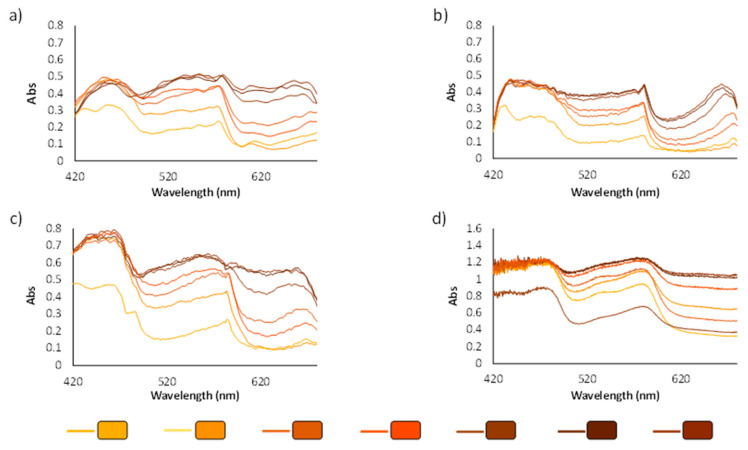
Spectra of the brown color group using different lights; (**a**) halogen lamp; (**b**) LED light; and (**c**) daylight compared to the (**d**) lab benchtop spectrophotometer.

**Figure 7 polymers-14-04285-f007:**
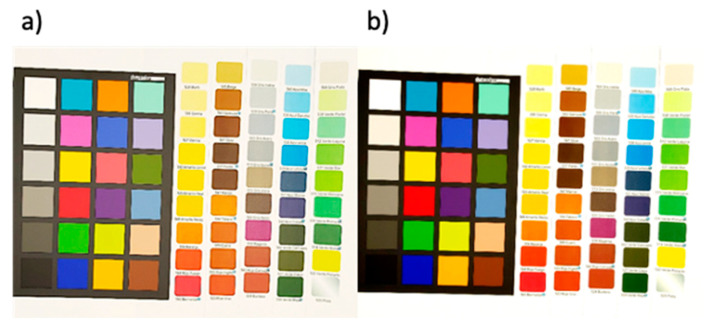
Comparison of the (**a**) original image and (**b**) processed image using the proposed procedure using an LED light source.

**Figure 8 polymers-14-04285-f008:**
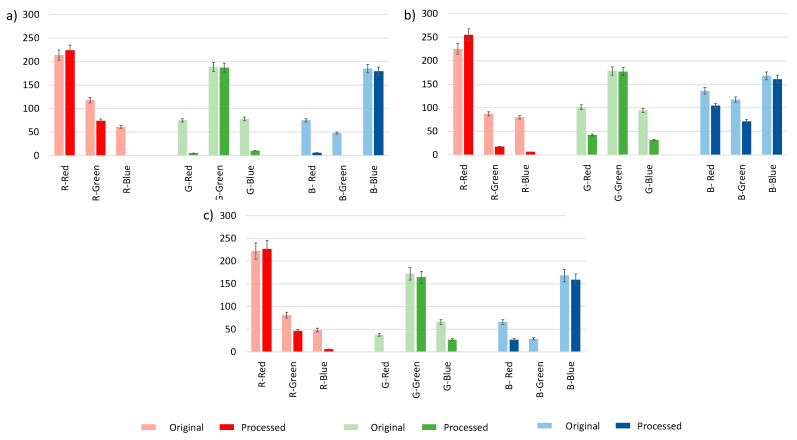
Obtained RGB coordinates with original and processed images from the three primary colors: Red, green and blue using different light sources: (**a**) Halogen, (**b**) LED and (**c**) daylight.

**Figure 9 polymers-14-04285-f009:**
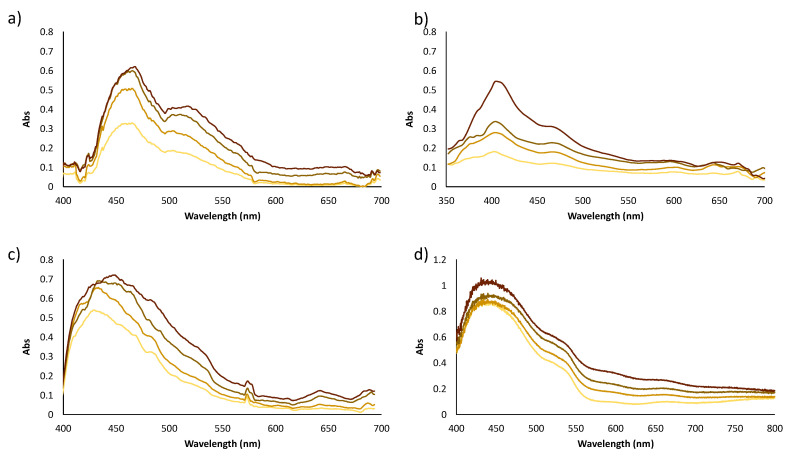
Comparison of the spectra obtained for different concentrations of ammonium in water using the PDMS solid sensor (0 mg·mL^−1^, 4 mg·mL^−1^, 8 mg·mL^−1^ and 12 mg·mL^−1^) for (**a**) the smartphone-mini-spectrometer in the transmission mode; (**b**) smartphone-mini-spectrometer in the reflection mode; (**c**) smartphone-mini-spectrometer coupled to a fiber optical probe reflection mode; and (**d**) a lab benchtop spectrophotometer.

**Table 1 polymers-14-04285-t001:** RSD values (%) of four colors (511, 529, 536 and 575) of the set using a smartphone spectrometer with different light sources and a lab benchtop spectrometer.

Light Source.	RSD (%) of 511 (λ = 475 nm)	RSD (%) of 529 (λ = 475 nm)	RSD (%) of 536 (λ = 660 nm)	RSD (%) of 575 (λ = 580 nm)
**Smartphone-Spectrometer**				
LED	2	2	1	0.6
Halogen	2	2	2	0.9
Daylight	4	3	2	3
**Laboratory spectrometer**				
UV-Vis	0.05	0.1	0.05	0.1

**Table 2 polymers-14-04285-t002:** Figures of merit of the sensors for urea and ammonium, expressed as ammonium, nitrite and hydrogen sulfide from GoSpectro. Transmission (T); Reflection (R). For further explanation, see the text below.

Sensor	RegisterMode	Light Source	Intercept (a ± s_a_)	Slope (mg^−1^ L) (b ± s_b_)	R^2^	Linearity Range (mgL^−1^)	LOD (mgL^−1^)
**Spectrometer me** **asurements**							
PDMS solid sensor (ammonium, urea) (490 nm)	T	Halogen	0.078 ± 0.004	0.0157 ± 0.0005	0.99	2.2–12	0.7
	R	Halogen	0.119 ± 0.003	0.0128 ± 0.0004	0.99	2.6–12	0.8
	Fiber optic probe, R	Halogen	0.07 ± 0.03	0.056 ± 0.005	0.97	0.6–8	0.17
PDMS delivery sensor (nitrite) (540 nm)	Fiber optic probe, R	Halogen	0.148 ± 0.013	0.52 ± 0.02	0.99	0.07–1.30	0.02
Cellulose solid sensor (hydrogen sulfide) (670 nm)	Fiber optic probe, R	Halogen	0.008 ± 0.005	0.041 ± 0.002	0.99	0.3–7	0.1
**RGB Coordinates measurements**							
PDMS solid sensor (ammonium/urea)	RGB image analysis	LED (G)	0.10 ± 0.02	0.104 ± 0.004	0.99	1.9–8	0.6
PDMS delivery sensor (nitrite)	RGB image analysis	LED (G)	0.10 ± 0.02	0.64 ± 0.02	0.99	0.02–2.70	0.01
Cellulose solid sensor (hidrogen sulfide)	RGB image analysis	LED (R)	0.054 ± 0.013	0.074 ± 0.002	0.99	1.8–13	0.5

## Data Availability

Not relevant.

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
