# Peer review of "Advances in the Measurement of Polymeric Colorimetric Sensors Using Portable Instrumentation: Testing the Light Influence"

_polymers, 2022, doi:10.3390/polym14204285_

Round 1

Reviewer 1 Report

The manuscript needs major revision before taking any decision. 

1. Extensive English correction is needed. There are many spelling mistakes. It is very difficult to follow. 

2. mention full form of PDMS in first line in abstract.

3. check the spelling in line 10, " imagens "

4. what is the meaning of " daily" in line 13 ?

5. expand "NQS " in line 17.

6. what do you mean by " These authors " in line 32?

7. expand " TEOS " in line 33.

8. I feel that it should be "ImageJ" in line 61.

9. check the lines 102-103 and 115-116. I feel that these are repeated. 

10. what kind of sensor in line 136 ?

11. what is the meaning to display various modes of detection fig 1 ?

Author Response

The manuscript needs major revision before taking any decision. 

  1. Extensive English correction is needed. There are many spelling mistakes. It is very difficult to follow.  According to your suggestions, English correction by a native with knowledge in chemistry has been made.
  2. mention full form of PDMS in first line in abstract. According to your suggestions, the full form of the abbreviation has been included.
  3. check the spelling in line 10, " imagens " Thank you it was a mistake
  4. what is the meaning of " daily" in line 13 ? Thank for you question- daily concert to the daylight
  5. expand "NQS " in line 17.OK
  6. what do you mean by " These authors " in line 32? We have avoided this confuse sentence. Please see new text.
  7. expand " TEOS " in line 33.OK
  8. I feel that it should be "ImageJ" in line 61. OK
  9. check the lines 102-103 and 115-116. I feel that these are repeated.  OK
  10. what kind of sensor in line 136 ? We have modified this paragraph for clarifying
  11. what is the meaning to display various modes of detection fig 1 ? We have improved this figure, new Figures 2 and 3 have been included. The text has been also modified in order to clarify the content of the paper.

Reviewer 2 Report

In my opinion manuscript polymers-1945825 is well written and deserves publication after revision.

Some suggestions to improve the manuscript:

1.       Give abbreviations in the abstract and in the text where first appear.

2.       Indicate novelty at the end of the abstract

3.       Revise English thoroughly: diary (line 11), daily (line 13) light should be daylight or day light; imagens should be images (line 10) easy use should be easy to use, (line 30), costumers should be customers (line 66), … . Avoid repetitions: line 19: results appears twice in the same sentence. The manuscript is hard to read, organize the information (especially the introduction), be more concise in the results section, use shorter phrases, …

4.       Explain originality at the end of the introduction.

5.       Part of subsection 3.1 should be included in the section 2 (lines 173 to 211). Fig. 3 may be included in the Results section

6.       Increase figures resolution and text labels size (especially the text in the Figures 2, 3 and 7 is almost illegible). Add statistical information where available (for example for the data given in fig. 3). These data should be better included in a table.

7.       Also move subsections 3.2 and 3.3 to section 2.

8.       Add error bars to data in Fig. 7.

9.       Compare your results with other data.

Author Response

In my opinion manuscript polymers-1945825 is well written and deserves publication after revision.

Some suggestions to improve the manuscript:

  1. Give abbreviations in the abstract and in the text where first appear. According to your suggestions, the full forms of the abbreviations have been included.
  2. Indicate novelty at the end of the abstract. The novelty of the article has been to show the influence of the light when color measurements are performed. Depending on the color of the sensors, the optimal light to be used is different.
  3. Revise English thoroughly: diary (line 11), daily (line 13) light should be daylight or day light; imagens should be images (line 10) easy use should be easy to use, (line 30), costumers should be customers (line 66), … . Avoid repetitions: line 19: results appears twice in the same sentence. The manuscript is hard to read, organize the information (especially the introduction), be more concise in the results section, use shorter phrases, … Thank you for your corrections. We have follow your suggestions.
  4. Explain originality at the end of the introduction. Thank you, a new paragraph has been included in the document.

  1. Part of subsection 3.1 should be included in the section 2 (lines 173 to 211). Fig. 3 may be included in the Results section. Thank you for your recommendations, according to your suggestions part of the subsection 3.1 has moved to section 2. Figure 3 has been included in the Results section.
  2. Increase figures resolution and text labels size (especially the text in the Figures 2, 3 and 7 is almost illegible). Add statistical information where available (for example for the data given in fig. 3). These data should be better included in a table. We have improved the figures and the text for clarifying the content of the paper in accordance with your statement
  3. Also move subsections 3.2 and 3.3 to section 2. Thank you for your suggestion, but we have not moved this subsections, because they include results of the research.
  4. Add error bars to data in Fig. 7. OK
  5. Compare your results with other data. We have compared the results with those obtained with lab benchtop spectrophotometer. See Figures 4, 5, 8 and S3 and S4 table 1,

Round 2

Reviewer 1 Report

The authors have addressed all the concerns.

It can be accepted in present form.